



# Air quality deterioration episode associated with typhoon over the complex topographic environment in central Taiwan

**Chuan-Yao Lin\*, Yang-Fan Sheng, Wan-Chin Chen, Charles, C. K. Chou, Yi-Yun Chien, Wen-Mei Chen**

Research Center for Environmental Changes, Academia Sinica, Taipei, Taiwan.

*Corresponding author

**Chuan Yao Lin,**

Research Center for Environmental Changes, Academia Sinica, Taipei, Taiwan

128 Sec. 2, Academia Rd, Nankang, Taipei 115, Taiwan

(E-mail: yao435@rcec.sinica.edu.tw, Tel.: +886-2-27875892, Fax: +886-2-27833584),



**Abstract:**
Air pollution is typically at its lowest in Taiwan during summer. The mean
concentrations of $PM_{10}$, $PM_{2.5}$, and daytime ozone (08:00–17:00 LST) during summer
(June–August) over central Taiwan are 35–40 µg/m3, 18–22 µg/m3, and 30–42 ppb,
respectively, between 2004 and 2019. Sampling analysis revealed that the contribution
of organic carbon (OC) in $PM_{2.5}$ could exceed 30% in urban and inland mountain sites
during July in 2017 and 2018. Frequent episodes of air quality deterioration occur over
the western plains of Taiwan when an easterly typhoon circulation interacts with the
complex topographic structure of the island. We explored an episode of air quality
deterioration that was associated with a typhoon between 15 and 17 July 2018, using the
Weather Research Forecasting with Chemistry (WRF-Chem) model. The results
indicated that the continual formation of low-pressure systems or typhoons in the area
between Taiwan and Luzon island in the Philippines provided a strong easterly ambient
flow, which lasted for an extended period between 15 and 17 July. The interaction
between the easterly flow and Taiwan's Central Mountain Range (CMR) resulted in
stable weather conditions and weak wind speed in western Taiwan during the study
period. Numerical modeling also indicated that a lee side vortex easily formation and
the wind direction could be changed from southwesterly to northwesterly over central
Taiwan because of the interaction between the typhoon circulation and the CMR. The



northwesterly wind coupled with a sea breeze was conducive to the transport of air
pollutants, from the coastal upstream industrial and urban areas to the inland area. The
dynamic process for the wind direction changed given a reasonable explanation why the
observed $SO_4^{2-}$ became the major contributor to $PM_{2.5}$ during the episode. $SO_4^{2-}$
contribution proportions (%) to $PM_{2.5}$ at the coastal, urban, and mountain sites were 9.4
µg/m3 (30.5%), 12.1 µg/m3 (29.9%), and 11.6 µg/m3 (29.7%), respectively. Moreover,
the variation of the boundary layer height had a strong effect on the concentration level
of both $PM_{2.5}$ and ozone. The combination of the lee vortex and land-sea breeze, as well
as the boundary layer development, were the key mechanisms in air pollutants
accumulation and transport. As typhoons frequently occur around Taiwan during
summer and fall, and their effect on the island's air quality merits further research
attention.



## 1. Introduction:

Tropical cyclones (also known as typhoons) are a frequent occurrence in East Asia
during summer and fall. Typhoons significantly affect not only meteorological parameters
but also air quality. That is because air pollution is strongly related to atmospheric
conditions, and typhoon circulation typically alters atmospheric stability and air pollutant
diffusion in specific locations. For example, researchers revealed that ozone episodes in
Hong Kong and southeastern China are strongly related to the passage of typhoons as
they approach the area (Lee et al., 2002; Ding et al., 2004; Huang et al., 2005 and 2006;
Yang et al., 2012; Zhang et al., 2013; Zhang et al., 2014; Wei et al., 2016; Yan et al. 2016;
Luo et al. 2018; Deng et al. 2019; Hung et al., 2021). The stagnant meteorological
conditions associated with strong subsidence and stable stratification in the boundary
layer results in pollutant accumulation before typhoons make landfall. Huang et al. (2005)
reported that approximately 30% of total ozone in Hong Kong was due to local chemical
production in the lower atmospheric boundary layer, and approximately 70% was
contributed by long-range transport from southern China (i.e., the Pearl River Delta).
According to the dynamic process perspective, Chow et al. (2018) reported frequent high-
O3 days when typhoons were located between Hong Kong and Taiwan (Fig. 1a) due to
the influence of the typhoon position and associated atmospheric circulations on air
quality.



58 Taiwan also experiences air quality deterioration as typhoons approach (Feng et al.

59 2007; Chang et al., 2011, Cheng et al., 2014; Hsu and Cheng, 2019). However, not all

60 typhoons are associated with poorer air quality in Taiwan. The effect of typhoons on air

61 quality is highly related to the location of the typhoon and its circulation's interaction

62 with Taiwan's Central Mountain Range (CMR; Fig. 1b). Thus, the mechanism of the

63 formation of poor air quality may differ between Taiwan and Hong Kong. Air quality

64 deterioration frequently occurs over the western plains of Taiwan when typhoons pass

65 between Taiwan and Luzon island in the Philippines; the distance of the typhoons from

66 Taiwan is typically several hundred kilometers but may even be greater than 1000

67 kilometers. Under such conditions, the weather is typically stable, with clear skies, strong

68 solar intensity, and weak wind speeds over Taiwan's western plains because of the

69 interactions of the typhoon's easterly circulations with the CMR. Furthermore, such

70 typhoons are usually associated with a Pacific high-pressure system during summer; thus,

71 the air temperature may be high. For example, researchers have noted that typhoon's

72 secondary circulation may enhance subsidence and result in a heat wave, clear skies, and

73 weak wind speed over Taiwan or Southern China (e.g., Ding et al. 2004; Huang et al.2005;

74 Jiang et al., 2015; Shu et al. 2016; Lam, et al., 2018) and thus adversely affect air quality

75 as well. In Taiwan, this phenomenon is particularly attributed to the blocking effect of the

76 CMR. The CMR occupies approximately two-thirds of Taiwan's landmass (300 km × 100



km) and lies NNE–SSW (Fig. 1b), with an average terrain height of approximately 2000
m (Lin and Chen, 2002; Lin et al., 2011) and some peaks of nearly 4000 m. The CMR
has a major effect on local circulation and interferes with the prevailing winds. When a
typhoon is located between Taiwan and Luzon, the low-level easterly airflow easily splits
northern and southern Taiwan and moves around the island, forming a vortex at the lee
side of the mountain (Hunt and Synder, 1980; Smolarkiewicz and Rotunno, 1989; Lin,
Y.L. 1993; Lin et al. 2007). On the leeside of the CMR, wind speeds are weak (Lin et al.,
2007) and the atmospheric conditions are more stable than on the windward side of
eastern Taiwan. Under these favorable conditions, air pollutants readily accumulate and
result in high ozone and aerosol concentrations over western Taiwan.

Summer and fall are regarded as "typhoon season" over Taiwan and throughout

East Asia. Statistically, more than 20 typhoons form in the western Pacific Ocean per year,
and approximately 3–4 typhoons directly strike Taiwan (Lin et al. 2011; Tu and Chen
2019). Records from Taiwan's Central Weather Bureau (CWB) indicate that 18% of
typhoons (Type 5; https://www.cwb.gov.tw/V8/C/K/Encyclopedia/typhoon/typhoon.pdf)
between 1911 and 2019 did not make landfall but passed between Taiwan and Luzon. The
wind circulations of this type of typhoon were easterly or southeasterly depending on the
location of the typhoons. Thus, it is not uncommon for more than 10 typhoons per year
to pass near Taiwan and affect the island's air quality. The impact of the interaction



between the typhoon's circulation and the CMR on the air quality on the lee side of the
mountain is more serious than in other areas.

To date, air pollution episodes with a formation mechanism associated with the

interactions between typhoon circulation and the CMR have not been thoroughly
documented in Taiwan. In this study, we investigated a major air quality event that
occurred on 17 July 2018, with a maximum $O_3$ concentration of 134 ppb and the daily
maximum aerosol concentration for $PM_{10}$ ($PM_{2.5}$) reaching 152 $\mu g/m^3$ (70 $\mu g/m^3$) in
inland rural areas of central Taiwan. We used the Weather Research Forecasting with
Chemistry model (WRF-Chem, version 3.9; Grell et al., 2005) to study the processes and
mechanisms of formation of the air pollution episode. The remainder of this paper is
organized as follows: Sect. 2 describes the data sources and sampling measurement during
the study period; Sect. 3 presents the model and settings used in this study; Sect. 4
presents the air quality characteristics and measurements recorded over the western plains
of Taiwan; Sect. 5 describes and discusses the simulation results of air quality associated
with the typhoon event using WRF-Chem; and finally, Sect. 6 provides the conclusions.

**2. Data sources and measurement**

We collected measurements of hourly $PM_{10}$, $PM_{2.5}$, and other pollutants ($O_3$, $NO_x$,

CO, and $SO_2$) as well as meteorological parameters (air temperature, wind field, and
rainfall) from Taiwan Environmental Protection Administration (TEPA) air quality



monitoring stations. To elucidate the spatial distribution of air pollutants, we classified
the observed stations over central Taiwan into "coast," "urban," and "mountain." Each
of these categories represents the mean concentration of the numbers derived from
stations of the same type. The coast category included two stations: Shalu (SL) and
Xianxi (XX; Fig. 1c). The urban category included five stations: Fengyuan (FY), Xitun
(XT), Zhongming (ZM), Changhua (CH), and Dali (DL; Fig. 1c). The mountain
category included three stations: Nantou (NT), Zhushan (ZS), and Puli (PL), which
were located nearby or in basins surrounded by high mountains (Fig. 1c). Two stations
on small islands were also considered in the analysis. One was in Kinmen (KM), which
is located close to Xiamen city in southeast China, and the other was Magong (MG)
station located in the Taiwan Strait (Fig. 1a).

To explore the air pollution episodes during summer, we recorded data in central

Taiwan in July 2017 and 2018. For the summer campaigns, we employed three
sampling sites (the squares in Fig. 1c), Shalu (SL, 24.23 °N, 120.57 °E; the same
location as the TEPA station), Zhushan (ZS, 23.76 °N, 120.68 °E; the same location as
the TEPA station), and Chung Shan Medical University (CSM) (24.12 °N, 120.65 °E;
Fig. 1c). ZS is a suburban site located in a complex valley surrounded by hills (300–
500 m) and high mountains (CMR; elevation > 2000 m) to the east and south,
respectively. The remaining two sampling sites, SL and CSM, were located in a coastal
suburban and urban area (Fig. 1c), respectively. The sampling period of each sample
was 12 h; daytime samples were collected from 08:00 to 19:00 LST, whereas nighttime
sampling was conducted from 20:00 LST to 07:00 LST. We determined mass
concentrations of the aerosols using a gravimetric measurement of the samples
collected on polytetrafluoroethylene membrane filters (Chou et al. 2008). Sounding
data (46734) were obtained from the CWB; the site on Penghu island was close to the



MG TEPA station (Fig. 1a).

During summer, the land-sea breeze easily combines with mountains' up/down

slope wind during daytime/nighttime. As the sea breeze develops, air flows are typically
transported from coastal areas and pass over the Taichung metropolitan region (Fig. 1c)
coupled with mountain slope flow to the inland area. The Taichung metropolis is a large
urban environment comprising residential, industrial, and agricultural lands (Cheng et
al., 2009). In particular, Taichung Power Plant (TPP, Fig. 1c), which is coal-fired, and
the Taichung Harbor Industrial (THI, Fig. 1c) zone are both located on the coast and
are responsible for substantial emissions in central Taiwan. Thus, severe emission
sources contribute to and affect the air quality in the Taichung metropolitan area under
favorable weather conditions. For detailed information on the instruments used in the
sampling analysis, please refer to Lee et al. (2019). Meteorological parameters,
including wind speed and wind direction, temperature, and relative humidity were
acquired from a meteorological station in the same location where data were collected
for this study.

**3. Model configurations**

In this study, we used the Weather Research and Forecasting model (WRF)

coupled with the WRF-Chem version 3.9 to study the air pollutants transport during
the episode. We obtained the meteorological initial and boundary conditions for
WRF-Chem from the National Center for Environmental Prediction (NCEP)
Operational Global Forecast system 0.25° × 0.25° data sets at 6-h intervals. We
selected the Yonsei University (YSU) planetary boundary layer (PBL) scheme for this



study. The coarse and fine domains had $259 \times 370$ and $301 \times 301$ grid nets with
resolutions of 9 km and 3km, respectively. The vertical had 41 levels, with the lowest
level approximately 40 m above the surface. To ensure that the meteorological fields
were well simulated, we employed the four-dimensional data assimilation scheme
according to the NCEP-GFS data. Transport processes included advection by winds,
convection by clouds, and diffusion by turbulent mixing. Removal processes included
gravitational settling, surface deposition, and wet deposition (scavenging in
convective updrafts and rainout or washout in large-scale precipitation). The kinetic
preprocessor (KPP) interface was used in both the chemistry scheme of the Regional
Atmospheric Chemistry Mechanism (Stockwell et al., 1990). The secondary organic
aerosol formation module, the Modal Aerosol Dynamics Model for Europe (MADE)
(Ackermann et al., 1998)/Volatility Basis Set (VBS) (Ahmadov et al., 2012) was
employed in the WRF-Chem model.

**4. Results and discussion:**
**4.1 Characteristics of air quality over central Taiwan**
Figures 2a–c indicate the monthly mean concentration for $PM_{10}$, $PM_{2.5}$, and
daytime (08:00–17:00 LST) ozone between 1994 and 2019. Clear seasonal variations
were noted for aerosol and ozone over central Taiwan. The lowest $PM_{10}$, $PM_{2.5}$, and
daytime ozone concentration were observed during summer (June–August) at 32–40
$\mu g/m^3$, 16–23 $\mu g/m^3$, and 35–42 ppb, respectively. The concentration of daytime ozone



peaked in October, whereas $PM_{10}$ and $PM_{2.5}$ peaked in March. In general, the highest
concentrations were observed in spring (March-May) and fall (September–November).
The daytime ozone peaked at 56 ppb and 48 ppb in October and April, respectively (Fig.
2c). For $PM_{10}$ and $PM_{2.5}$, the peak concentrations were 70–75 µg/m$^3$ and 40–45 µg/m$^3$
over the western plains in March (Fig. 2a, b). Regarding the characteristics of ozone
distribution, the concentration at the mountain site was typically higher than that in
urban areas and the coast. For $PM_{10}$ and $PM_{2.5}$, the mountain site also typically had
higher concentrations than did the urban and coastal areas, except during summer (Fig.
2a,b). The monsoon dominates the prevailing wind over East Asia. During summer, a
southwesterly wind prevails, whereas a northeasterly wind prevails during fall, winter,
and spring. The characteristics of the seasonal variations might be due to the summer
having a cleaner background and higher boundary layer height than those in other
seasons. As mentioned earlier, the major emission sources such as industry and traffic
are located in coastal and urban areas. The mean highest concentration of ozone
typically occurs over rural mountain areas during summer; thus, the dominant land-sea
breeze might play a critical role in the air quality in western Taiwan.

During summer (July only in this study) in 2017 and 2018, we conducted sampling

campaigns in central Taiwan. Table 1 presents the mean concentration of the elements
in $PM_{2.5}$ at sampling stations SL, CSM, and ZS during July in 2017 and 2018. The mean
concentration of $PM_{2.5}$ for stations SL, CSM, and ZS were 15.7, 16.9, and 21.4 µg/m3.
The inland rural mountain site, ZS, clearly had the highest total $PM_{2.5}$ concentration.
Organic carbon (OC) and $SO_4^{2-}$ had the highest concentrations of the species in $PM_{2.5}$,
and both increased from the coast to the inland mountain area (Table 1). Because the
major emissions were from coastal industry or urban areas, sea breeze transport played
a role in $PM_{2.5}$ concentration in the western plain. The major contributing species in



PM$_{2.5}$ were OC, SO$_4^{2-}$, NO$_3^-$, NH$_4^+$, and elemental carbon (EC; Table 1). At the coastal
station SL, the concentrations of OC and SO$_4^{2-}$ were comparable at 4.3 µg/m$^3$ and 4.5
µg/m$^3$, accounting for 27.5% and 28.6% of PM$_{2.5}$, respectively. At the city site CSM
and the inland rural mountain station ZS, OC had concentrations of 5.6 (33.1% of PM$_{2.5}$)
and 6.6 µg/m$^3$ (30.9% of PM$_{2.5}$), respectively. The results indicated that the contribution
of OC in PM$_{2.5}$ could exceed 30% at the urban and inland mountain sites. The
concentration of OC increased from the coast (4.3 µg/m$^3$; 27.5% of PM$_{2.5}$) to the
mountain station (6.6 µg/m$^3$; 30.9% of PM$_{2.5}$), and the urban site had the highest
proportion (5.6 µg/m$^3$; 33.1% of PM$_{2.5}$) in PM$_{2.5}$ among these stations (Table 1). SO$_4^{2-}$
also exhibited an increased concentration from coastal areas to the inland mountain area,
but the changes were minor (4.5–4.8 µg/m$^3$). Notably, the proportion of SO$_4^{2-}$ in PM$_{2.5}$
decreased from the coast to the mountain area because the major sources, TPP and THI
(Fig. 1c), are located on the coast. The other species, namely NO$_3^-$, NH$_4^+$, and EC, at
SL, CSM, and ZS had comparable concentrations between stations (1.0–1.4, 1.7–2.0,
and 1.1–1.4 µg/m3, respectively; Table 1). The inland rural station ZS was located in a
foothill valley of the CMR and surrounded by mountains. Thus, the high concentration
at ZS might be due to sea breeze transport.

In general, OC and SO$_4^{2-}$ were the major species over western Taiwan, especially

in inland areas. These results suggest that local contribution, such as traffic, industry,
and even agricultural emissions, might play critical roles in the composition of PM$_{2.5}$.
Furthermore, the spatial distributions of highest PM$_{2.5}$ and daytime ozone concentration
were not always in urban areas; instead, concentrations accumulated in inland rural
areas (Fig. 2 and Table 1). The roles that the land-sea breeze, boundary layer
development, and interaction of typhoon circulation with complex geographic
structures play in air quality require clarification. The mechanism of these complex





processes and local circulation variations are demonstrated through a case study using
numerical model simulation in Sect. 4.2.2.

## 4.2 Air quality deterioration case from 15–17 July 2018

### 4.2.1 Weather condition and observation

To explore air quality deterioration processes and formation mechanisms, we
employed a severe air pollution episode between 15 and 17 July 2018. Weather maps
obtained from the NCEP Global Forecast System (GFS) revealed that a tropical
depression formed to the east of the Philippines and moved northwestward on 15 July
2018 (Fig. 3a). Another low-pressure system followed, located to the south of this
tropical depression on 16 July (Fig. 3b). On 17 July, this tropical depression
strengthened and formed a weak typhoon named SONTINH, located between Taiwan
and Luzon island in the Philippines (Fig. 3c); the original low-pressure system also
strengthened into a tropical depression on 17 July. The continual formation of low-
pressure systems or typhoons to the east of Luzon shifted the ambient wind flow of
Taiwan to an easterly direction for an extended period between 15 and 17 July (Fig. 3a-
c). The easterly ambient flow was easily blocked by Taiwan's CMR, resulting in a lee
vortex formation associated with stable atmospheric conditions and weak wind speed
in western Taiwan. The mechanism of lee vortex formation on the lee side of a high
mountain has been described through a laboratory experiment (Hunt and Synder, 1980)
and numerical modeling (e.g., Smolarkiewicz and Rotunno, 1989). Li and Chen (1998)



employed a wind flow with low Froude number (<0.5) (Fr ≡ U/NH, where U is the far
upstream flow speed; N is the Brunt–Vaisala frequency, a measure of stratification; and
H is the height of an obstacle), and the low-level airflow easily split off the northern
coast and moved around the island of Taiwan. The current study is an example of a low
Fr case (<0.5; assumed average wind speed, U = 10 ms$^{-1}$; Brunt–Vaisala frequency, N
= 10$^{-2}$s$^{-1}$; and average mountain height, H = 2.5 km). Thus, we expected wind speeds to
be weak and atmospheric conditions to be more stable on the lee side of the CMR
compared with the windward side of eastern Taiwan.

Sounding data (Fig. 4) recorded at the CWB station in Penghu island (46734,

close to MG in Fig. 1a) indicated a relatively weak wind speed (<5 m/s) in the low
boundary (below 850 hPa) during the study period from 15 to 17 July 2018 (Fig.4a-c).
Above 700 hPa (3000 m), a strong easterly wind (>10 m/s) prevailed due to the typhoon
circulations. Furthermore, clear subsidence and multiple inversion layers were revealed
in the sounding between 16 and 17 July (Fig. 4b,c). On 17 July, the inversion layer was
even lower than 950 hPa (Fig. 4c); that is, only a few hundred meters over Penghu
island in the Taiwan Strait. The sounding data revealed stable atmospheric conditions,
high relative humidity, and weak wind speed on the leeside of the mountains over
western Taiwan.

Figure 5 displays the variations in wind field and air pollutants (both PM$_{2.5}$ and





ozone) at the TEPA stations on two small islands, KM and MG (locations marked in
Fig. 1a) and results over the western plain from 12 to 18 July 2018. The wind direction
and wind speed were quite different between these two stations and over the western
plains (Fig. 5a). The wind speed was relatively strong at KM, especially between 16
and 17 July because the typhoon circulation had already reached the coastal area of
China and the Taiwan Strait. The wind direction was originally southerly on 12 July,
becoming northeasterly after 12:00 LST on 14 July 2018. During periods of strong wind
speed at KM, the concentrations of $PM_{2.5}$ and $O_3$ revealed no diurnal variation and a
steady low, with $PM_{2.5} < 15$ µg/m$^3$ and daytime $O_3 < 40$ ppb after 12:00 LST on 14 July.
The wind speed at MG was weaker than that at KM because MG is close to Taiwan and
was likely affected by the mountain blocking effect mentioned earlier. Because the wind
speed did not change considerably, the $PM_{2.5}$ and $O_3$ concentration levels did not
fluctuate obviously at MG during the study period.

By contrast, the wind field time series indicated clear land–sea breeze variations

over western Taiwan. At the inland mountain site, wind speed was relatively weak
compared with the coastal and urban sites (Fig. 5a). The $PM_{2.5}$ and ozone time series
for the coastal, urban, mountain sites are presented in Fig. 5b-c. The $PM_{2.5}$
concentrations at the urban and mountain sites ranged from 30 to 60 ug/m$^3$ between 16
and 17 July 2018. Notably, the timing of peak $PM_{2.5}$ concentration differed between the



coastal, urban, and mountain sites. Peak $PM_{2.5}$ at the coastal and urban sites was
observed around noon, whereas peak $PM_{2.5}$ at the inland mountain site occurred at 18:00
LT on 17 July 2018 (Fig. 5b). The differences in the timing of the peak $PM_{2.5}$
concentrations between the coastal and urban sites and the inland mountain site could
be attributed to the transport of the sea breeze. No clear diurnal variation in $PM_{2.5}$
concentration was observed between the urban and mountain sites between 16 and 17
July. That is, even at night and in the early morning, the concentration remained as high
as 40 $\mu g/m^3$ (Fig. 5b) because atmospheric conditions were favorable for air pollutant
accumulation. The peak ozone concentration occurred around noon at the coast and
urban sites, whereas the peak at the mountain site occurred later at 16:00 LST (Fig. 5c).
We estimated that the concentrations of $PM_{2.5}$ and ozone on the episode day on 17 July
(Fig. 5b,c) were three times higher than the mean concentration during summer (Fig. 2)
in central Taiwan. As mentioned earlier, the major emissions were generated by coastal
industry and the Taichung city metropolitan area, but the peak ozone concentration
occurred at the inland mountain station (120 ppb at PL) because of sea breeze transport
from upstream to downstream sites.
Spatial distribution of wind field and $PM_{2.5}$ concentration (Fig. 6) from TEPA
stations in Taiwan revealed a strong easterly wind in northern and southern Taiwan and
weak wind speed and clear sea breeze development during daytime in central Taiwan.



PM$_{2.5}$ concentrations remained low (<15 µg/m$^3$) at the northern, eastern, and southern
tips of Taiwan on 17 July 2018 (Fig. 6a-f). Over western Taiwan, a sea breeze developed
after 10:00 LST, and a strong onshore flow blew air pollutants to the inland area(Fig.6b-
d). A high PM$_{2.5}$ concentration (>50 µg/m$^3$) extended from the coast to the urban area
at noon (Fig. 6b-c), which was subsequently transported to the inland mountain area in
the afternoon and nighttime (Fig. 6d-f). The high PM$_{2.5}$ concentration accumulated in
Maoli county (located north of Taichung city) at midnight owing to the convergence of
southerly and land breeze (Fig. 6f). Actually, the spatial variation of PM$_{2.5}$ could also
be observed on the previous day (16 July; Fig. 5b), which contributed approximately
30 µg/m$^3$ in the early morning on 17 July in central Taiwan.

The location of the high-pollution ozone was also strongly associated with the

land-sea breeze during the daytime (Fig. 7 b-e). A high concentration of ozone was
observed at the urban station at noontime (Fig. 7c); the ozone was transported to the
inland mountain station, resulting in peak concentrations higher than 120 ppb between
16:00 and 18:00 LST (Fig. 7d-f). By 22:00 LST, the ozone concentration had declined
more rapidly in the city than in the mountain area because of the dilution effect (Fig. 7
g-h). The detailed pollution process and mechanism are demonstrated and discussed in
the model simulation in Sect. 4.2.2.
**4.2.2 Simulation Results:**



The hourly comparison between observed (red solid) and simulated (blue dashed)
$PM_{2.5}$ and ozone between 12 and 18 July 2018 are presented in Fig. 5b,c. In general,
our simulation reasonably captured the variation of $PM_{2.5}$ and ozone in western Taiwan
and small island sites, MG and KM (Table 2). For $PM_{2.5}$, the root mean square error
(RMSE) at all sites was less than 1.0 $\mu g/m^3$, and the correlation between observed and
simulated values was 0.72 and 0.81 at the urban and mountain sites, respectively.
Regarding the mean bias of $PM_{2.5}$, it was slightly overestimated at coastal and urban
sites and underestimated at the mountain site and sites on the two islands. In the ozone
simulation, the correlation between observed and simulated values was as high as 0.73–
0.9, except for MG. The RMSE of ozone for all areas was less than 1.45 ppb. For the
mean bias of ozone, the maximum underestimation (−10 ppb) occurred at the coastal
site, and the maximum overestimation (13.8 ppb) occurred over the mountain area
because of the simulation of the spatial distribution difference.
Figure 8 indicates the simulated wind field (streamline) and spatial distribution of
$PM_{2.5}$ on 17 July 2018. The ambient wind flow was easterly and blocked by the CMR;
the wind flow went around the CMR during the study period. The strongest wind speeds
were recorded at the northern and southern tips of Taiwan and the coastal area of
southeastern China (Fig. 8). By contrast, the wind speed was relatively weak on the lee
side of the CMR from the middle of the Taiwan Strait to western Taiwan. This finding



is consistent with the observed wind speed being stronger at KM (Fig. 5a) than in the
area over western Taiwan. Figure 8a–f reveals that the highest $PM_{2.5}$ concentration (>60
µg/m3) occurred on the lee side of the CMR in central Taiwan during the daytime
(08:00–16:00 LST) on 17 July 2018. After 08:00 LST, the sea breeze gradually
developed and the onshore wind speed increased (Fig. 8a-c); thus, the high-
concentration $PM_{2.5}$ plume was transported from the coast to the inland mountain area.
Even though the area has high emissions, the $PM_{2.5}$ concentration along the coastal area
of China was low because of the strong wind speed (Fig. 8a-c). As sea breeze developed
after 08:00 LST, and the vortex circulation was coupled with the onshore flow (Fig. 8a–
d). The lee vortex circulation was not clear because it combined with the sea breeze and
enhanced the air pollutant transport to the inland area during the daytime. However, the
lee vortex circulation was clearly formed in the area from 23.5 to 24.5 °N in the
afternoon until early morning on the next day because the land breeze interacted with
the mountain lee-side flows (Fig. 8e-f). After the lee vortex circulation formed, the
southerly flow in the western plain was enhanced (Fig. 8e-f). These processes resulted
in trapped air pollutants over the plain area because of the interaction between the lee
vortex southerly component wind and the offshore flow in the nighttime and early
morning. This also explains the absence of diurnal $PM_{2.5}$ variation and high
concentration (>35 µg/m$^3$) accumulated during nighttime and early morning on July 16





and 17 over central Taiwan (Figs. 5a,b, and 6f). Thus, the lee vortex formation was
adverse to the development of the offshore flow (land breeze) and prolonged the air
pollutant accumulation in western central Taiwan (Fig. 6 and 8). These critical
processes explain why air pollutants tended to accumulate in central Taiwan during the
episode days. Notably, the wind speed was strong and the concentration of $PM_{2.5}$ was
low in the Taiwan Strait close to coastal areas of China in the simulation (Fig. 8a-f) and
according to observations at KM (Fig. 5a). According to the spatial distribution, a strong
wind speed can limit the number of air pollutants transported southward from mainland
China to Taiwan (Fig. 8b-f). That is, the pollution type was locally dominated during
the event days.

Similar to the observed zone (Fig.7), the simulated ozone (Fig.9) was also

dominated by circulations associated with the land-sea breeze and the interaction of the
easterly flow with the CMR. Most of the area had steady low concentrations in the early
morning on 17 July (Fig. 9a) because of the dilution effect of the ozone formation in
the nighttime and early morning (Fig. 9a and h-i). A high concentration already existed
over the mountain area in Miaoli County (Fig.1b) in the early morning at 04:00 LST
(Fig. 9a), with a steady low concentration over the coastal and urban areas. During the
daytime, the background ozone concentration was 25–35 ppb over the ocean. The ozone
concentration promptly increased around noon and extended over almost the entire



western plains in the afternoon (Fig.9 c-f) on 17 July. The area of high ozone
concentration extended over the western plains when the sea breeze developed after
10:00 LST on 17 July (Fig. 9c). Following increases in wind speed, the high ozone
concentration extended to the inland area and was transported further south of Taichung
City (Fig. 9 d–e). The peak ozone concentration at the inland rural site occurred at 16:00
LST, whereas it occurred in the city center at the urban site at 12-14:00 LST (Figs. 5c;
7c,d; 9d,e). Because the major emission sources were coastal industry and the urban
area, the high ozone concentration at the inland site was the result of ozone being
transported by the sea breeze. The simulated peak ozone concentration occurred
between 14:00 and 16:00 LST at the inland site because of the sea breeze coupled with
the mountain upslope wind (Fig.9 c–f). Moreover, the high-ozone plume was associated
with the lee vortex circulation over the Taiwan Strait and provided a southerly flow
component during the nighttime and early morning (Fig. 9a, and g–i).

As mentioned earlier, sounding data indicated multiple inversion layers on the

event days. To further investigate the boundary layer development and air pollutant
distribution in the vertical, a northwest-southeast cross-section AA' (Fig.10a) was
superimposed over the high concentration area, as illustrated in Fig. 10. In the early
morning at 05 LST (Fig.10b), a separate high-concentration plume was observed at
ground level and another remained at an elevation of 1000 m on 17 July. It is a typical



boundary layer structure due to ground surface radiation cooling under stable
atmospheric conditions during nighttime and early morning. These two layers' plume
coupled together due to boundary layer gradually developed in the morning after 0700
LST(Fig.10 b-d). Because the emissions increased during rush hour, the concentration
promptly increased as the $PM_{2.5}$ plumes of these two layers coupled well in the vertical
below 1000 m at 10:00 LST (Fig. 10 d). The wind speed was weak at elevations below
1500 m but strong and offshore in a southeast-northwest direction above 2000 m due to
easterly tropical cyclone circulation. The high-PM2.5 plume (concentration > 50 $\mu g/m^3$)
was pushed by the sea breeze coupled with the upslope wind and accumulated in the
inland rural area during daytime (12:00–16:00 LST) (Fig. 10e–g). The highest
concentration was not at ground level but heights between 500 and 1000 m at noontime
(Fig.10e) and 1000-1500 m in the afternoon (Fig. 10 f-g). The boundary layer structure
and the coupled between sea breeze and mountain upslope wind played important roles
for the $PM_{2.5}$ concentration distribution in the vertical along the cross-section (Fig.10d-
g). As offshore wind developed, which pushed the air pollutants from the mountain area
to the plain and coastal area (Fig. 10 g–i), and the elevation of the plume was
predominantly between 500 and 1500 m after 20:00 LST. The discussion above
indicated that $PM_{2.5}$ concentration was not only strongly related to the interaction of
ambient flow with the CMR but also the diurnal variations in boundary layer



development.

Figure 11 indicates the ozone cross-section in a northwest-southeast direction in

Fig. 10a. A low ozone concentration (<25 ppb) was observed near ground level because
of the dilution effect in the early morning at 04:00 LST (Fig. 11a) on 17 July. However,
a high-ozone layer was observed between 500 and 1500 m because of the previous
day's contribution. After 08:00 LST, the mixing layer developed, and emissions from
traffic and industry also increased. Concurrently, both the onshore sea breeze over the
plain and the upslope wind over the mountain developed; thus, wind speed also
enhanced in the low boundary (Fig. 11b-e). The sea breeze and weak wind speed also
exacerbated the high-concentration ozone in the inland area during the daytime (Fig.
11c–f). At nighttime, the ozone concentration gradually decreased because of the
dilution effect below 500 m (Fig. 11h-i). However, TEPA measurements revealed that
a layer with high ozone concentration remained between 1000 and 1500 m (Fig. 7g-h)
because low $NO_x$ was emitted over the mountain area in Taichung and Miaoli county.
This also explains why the high ozone concentration first occurred over the mountain
slope area as a result of the concurrent sea breeze and upslope wind in the morning
(Figs. 9a and 11a). That is, the area of high concentration occurred earlier in the low-
emission mountain area than on the plains, a major emission area. The simulated ozone
concentration indicated that the high concentration did not occur near ground level but



at 800–1000 m. This phenomenon was closely related to the development of the
boundary layer structure and its interaction with the upper residual layer formation on
the previous day.
**5.  Discussion:**

The wind direction over Taiwan during summer is mostly southerly to

southwesterly (Table 1). However, the wind direction during the episode was westerly
to northwesterly (Table 2). The wind direction changed because of the critical
interaction between typhoon circulations and the CMR. Moreover, the concentration of
$PM_{2.5}$ and its composition during the episode also differed significantly from the
monthly mean, as revealed in Table 2. A substantial increase in daily mean $PM_{2.5}$ was
observed at all sites, especially at the CSM site (urban), where concentration increased
from 16.9 to 40.5 µg/m$^3$ (Table 2). Furthermore, $SO_4^{2-}$ became the dominant species in
$PM_{2.5}$ from the coastal to the mountain area, ranging from 30.5 to 29.7% during the
episode. The $SO_4^{2-}$ concentration during the episode (Table 2) was more than twice that
of the monthly mean (Table 1) in the Taichung area. This variation was due to the wind
direction changing from southwesterly to northwesterly, resulting in a contribution
increase from the upstream TPP and THI (Fig. 1c), which are the major sources in
central Taiwan.

On 17 July 2018, Taichung City not only experienced high air pollutant



concentrations but also a maximum air temperature as high as 35.4 °C. That is, a heat
wave (Lin et al., 2017; Kuth et al. 2017) occurred on 17 July because of the subsidence
of the typhoon circulation on the lee side of the mountain. The daily mean temperature
for the sampling sites between 15 and 17 July for SL, CSM, and ZS were 29.9 °C, 30
°C, and 29.4 °C, respectively. However, the monthly mean temperatures (July in 2017
and 2018) during the sampling period for SL, CSM, and ZS were 28.9°C, 28.8°C, and
26.5°C, respectively. Thus, the daily mean temperature during the episode period was
1–2 °C higher than is typical for days in July. In general, the mean wind speed on the
episode days at these three sites was weaker (<1 m/s) than the monthly mean (Tables 1
and 2). Such stable weather conditions, weak wind speed, and high air temperature were
conducive to the generation and formation of a secondary aerosol. This is exemplified
by the concentrations of other species, such as OC, $NO_3^-$ and $NH_4^+$, being considerably
higher during the episode days (Table 2) compared with the monthly mean in Table 1.
Notably, EC increased to a lesser extent than did the other species. These results suggest
that secondary aerosol plays a critical role under such stable weather conditions and
wind direction. Because ambient wind changes during typhoon formation between
Taiwan and Luzon island in the Philippines are not uncommon, the air quality impacts
in such weather conditions merit further research. A detailed discussion of variations in
aerosol chemical composition transformation will be presented in a separate paper.



## 6. Summary:

The lowest air pollution levels in Taiwan typically occur during summer because of a low air pollution background under southwesterly prevailing winds and the higher boundary elevation associated with high air temperatures. The monthly mean concentrations of $PM_{10}$, $PM_{2.5}$, and daytime ozone (08:00–17:00 LT) in summer (June–August) during 2004-2019 over central Taiwan are 35–40 $\mu g/m^3$, 18–22 $\mu g/m^3$, and 30–42 ppb, respectively. Sampling analysis also indicated that the contribution of OC in $PM_{2.5}$ could exceed 30% in urban and inland mountain sites. However, episodes of poor air quality frequently occur over the western plains when an easterly typhoon circulation interacts with the complex topographic structure in Taiwan. Under such a weather condition, concentrations of $PM_{2.5}$ and ozone could be higher than 2 times of those monthly mean. During the episode, $SO_4^{2-}$ became the major contributor to $PM_{2.5}$, and its concentration and contribution proportion (%) in $PM_{2.5}$ at coastal, urban, and mountain sites were 9.4 $\mu g/m^3$ (30.5%), 12.1 $\mu g/m^3$ (29.9%), and 11.6 $\mu g/m^3$ (29.7%), respectively. It is due to the northwesterly wind was conducive to the transport of $SO_2$ and sulfate from the coastal upstream major emission sources (areas in TPP and THI) to the inland area.

To explore the mechanism of air pollution formation, we conducted a detailed data analysis and WRF-chem model simulation of an episode of poor air quality



between 15 and 17 July 2018. Numerical modeling indicated that not only wind
direction changes due to lee vortex but also boundary layer development were the key
mechanisms in the transport of air pollutants. Typhoons are a frequent occurrence in the
area around Taiwan during summer and fall.   Because of Taiwan's complex
geographic structure, the flow patterns and diurnal boundary layer variations resulted
in the high concentrations of ozone and $PM_{2.5}$ and composition of $PM_{2.5}$ during the
episode deviating from the monthly mean in summer. The results of this study
contribute valuable data on the effect of extreme weather, such as typhoon circulation,
on the weather parameters and air quality in Taiwan. We summarize the key
mechanisms and processes of the interaction between, typhoon circulation, lee vortex,
land-sea breeze, boundary layer development, and topography and their effects on air
quality in Fig. 12.
(1) First, typhoon circulations provided a strong easterly ambient flow. This easterly

flow interacted with the CMR, resulting in a lee vortex formation over western

Taiwan. (Fig.12, left panel)

(2) During the nighttime, the offshore wind that developed pushed the air pollutants

from the mountain area to the plain and coastal areas. Concurrently, a clear lee

vortex formation could be observed near Taiwan's coastal area in the Taiwan

Strait and thus a southerly flow in the western plains was enhanced. These



processes resulted in trapped air pollutants over the Taichung area and the
mountain area in Miaoli county (Fig. 1b) in western Taiwan. The boundary layer
height was low because of ground surface radiation cooling and inversion layer
formation. The air pollutants remained separate because of considerable
decreases in emissions at ground level coupled with the boundary residual layer
being at a higher elevation. (Fig.12, right top panel)
(3) In the morning, this residual layer with polluted air mass combined with and
contributed to the ground surface air concentration level because the boundary
layer height increased. This also explains why the ozone and $PM_{2.5}$ concentrations
dramatically increased after the boundary layer development during the daytime.
For this reason, the high-concentration ozone plume was located in a low-
emission mountain area and the episode occurred at an earlier time than in the
plain area where the major emission sources are located.
During the daytime, the lee vortex flow coupled with a sea breeze and combined
with a mountain upslope wind; this resulted in the accumulation of air pollutants
in the inland mountain area. Furthermore, because of the mountain upslope flow,
the high $PM_{2.5}$ and ozone concentrations were located not at ground level but at
heights between 500 and 1000 m. The peak concentration at the inland mountain
site occurred approximately 4–6 hours later than at the upstream coastal site


because of the sea breeze. (Fig.12 right down panel)
**Acknowledgements:**
The accomplishment of this work has financial support from the Ministry of Science
and Technology, Taiwan, under grants 108-2111-M-001-002 and 109-2111-M-001-004.

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






Table 1: Concentrations of PM2.5 and its major components and daytime ozone as
well as meteorological parameters at the SL (coastal), CSM (urban), and ZS
(mountain) sampling sites in July 2017 and 2018.

| | 201707 201807 | | | | | |
|---|---|---|---|---|---|---|
| | Coast (SL) | | Urban (CSM) | | Mountain (ZS) | |
| | Value | (%) | Value | (%) | Value | (%) |
| PM$_{2.5}$ (µg/m$^3$) | 15.7 | | 16.9 | | 21.4 | |
| SO4$^{2-}$ (µg/m$^3$) | 4.5 | (28.6%) | 4.6 | (27.5%) | 4.8 | (22.2%) |
| OC (µg/m$^3$) | 4.3 | (27.5%) | 5.6 | (33.1%) | 6.6 | (30.9%) |
| NO3$^-$ (µg/m$^3$) | 1.4 | (9.1%) | 1.0 | (6.0%) | 1.1 | (5.3%) |
| NH4$^+$ (µg/m$^3$) | 1.7 | (10.5%) | 1.7 | (9.9%) | 2.0 | (9.3%) |
| EC (µg/m$^3$) | 1.1 | (6.7%) | 1.1 | (6.3%) | 1.4 | (6.4%) |
| O$_3$ (ppb, 08-17LST) | 35.3 | | 39.4 | | 39.7 | |
| T ( °C ) | 28.9 | | 28.8 | | 26.5 | |
| ws ( m/s ) | 0.7 | | 0.9 | | 0.5 | |
| wd ( ° ) | 207.7 | | 238.4 | | 247.5 | |


Table 2: Concentrations of PM2.5 and its major components and daytime ozone as
well as meteorological parameters at the SL (coastal), CSM (urban), and ZS
(mountain) sampling sites between 15 and 17 July 2018.

| | 2018/07/15-2018/07/17 | | | | | |
|---|---|---|---|---|---|---|
| | Coast (SL) | | Urban (CSM) | | Mountain (ZS) | |
| | Value | (%) | Value | (%) | Value | (%) |
| PM$_{2.5}$ (µg/m$^3$) | 30.9 | | 40.5 | | 39.2 | |
| SO4$^{2-}$ (µg/m$^3$) | 9.4 | (30.5%) | 12.1 | (29.9%) | 11.6 | (29.7%) |
| OC (µg/m$^3$) | 6.9 | (22.2%) | 9.7 | (23.8%) | 8.1 | (20.7%) |
| NO3$^-$ (µg/m$^3$) | 2.9 | (9.5%) | 2.9 | (7.0%) | 4.4 | (11.1%) |
| NH4$^+$ (µg/m$^3$) | 4.0 | (12.9%) | 4.7 | (11.7%) | 5.9 | (15.0%) |
| EC (µg/m$^3$) | 1.8 | (6.0%) | 1.5 | (3.6%) | 2.0 | (5.2%) |
| O$_3$ (ppb, 08-17LST) | 65.5 | | 74.1 | | 64.7 | |
| T ( °C ) | 29.9 | | 30.0 | | 29.4 | |
| ws ( m/s ) | 0.6 | | 0.9 | | 0.8 | |
| wd ( ° ) | 290.6 | | 250.7 | | 279.9 | |









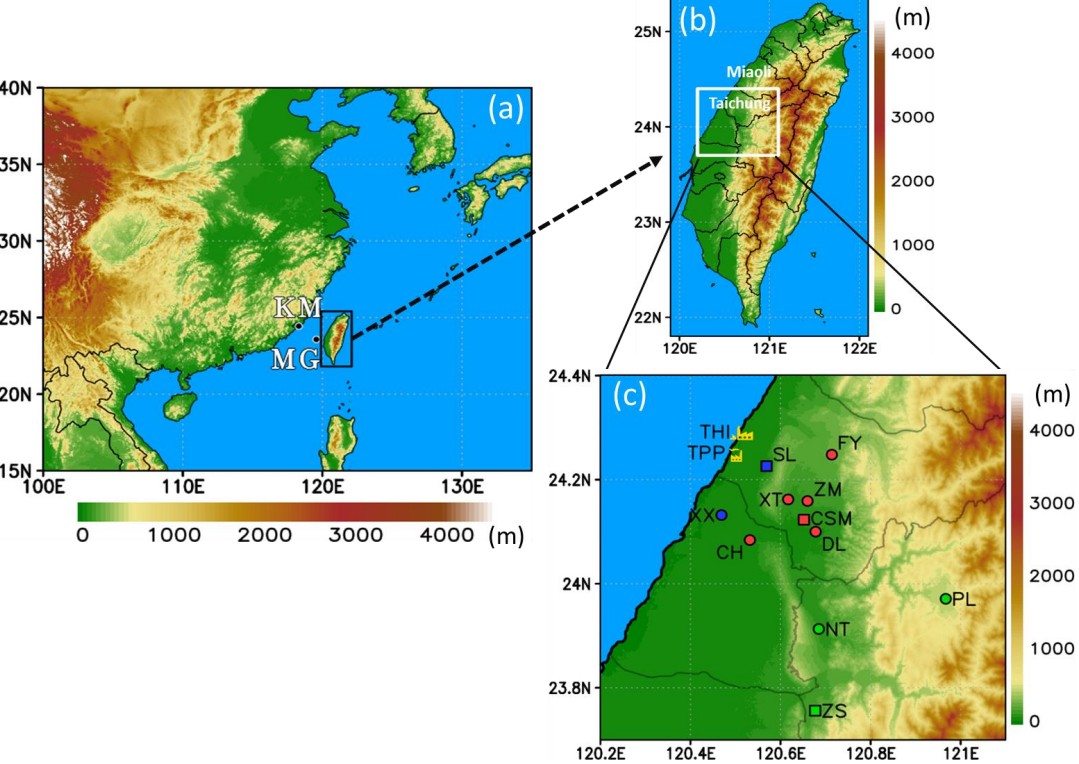



Fig. 1. (a) Location of Taiwan and surrounding countries in East Asia. KM and MG
are the island stations of the Taiwan Environmental Protection Administration
(TEPA). (b) Topography over Taiwan and the locations of Taichung city and Miaoli
county. (C) Location of TEPA air quality monitoring stations in central Taiwan in
coastal (SL and XX), urban (FY, XT, ZM, CH, and DL), and mountain (PL, NT, and
ZS) areas. TPP, Taichung Power Plant; THI, Taichung Harbor Industrial area.




(a)

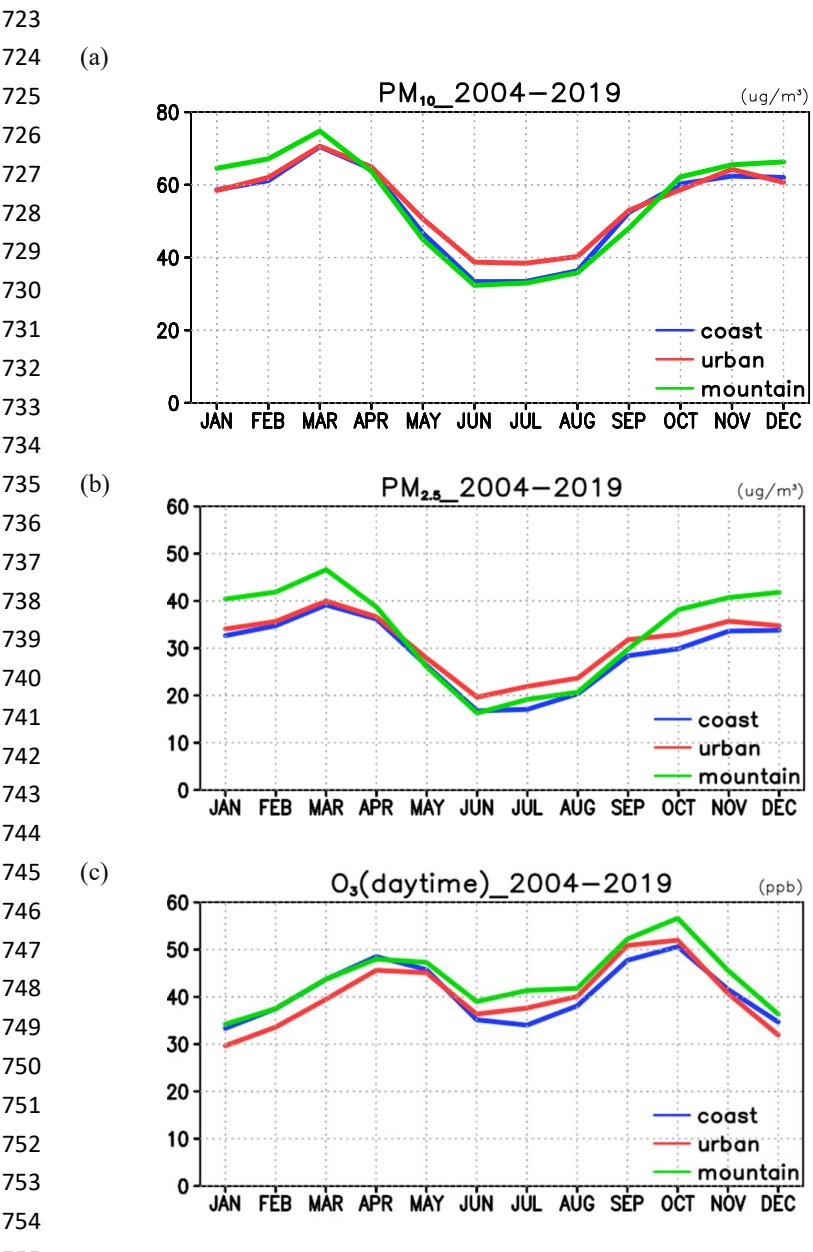

Fig. 2. Average monthly concentrations of (a) $PM_{10}$, (b) $PM_{2.5}$, and (c) daytime
(08:00–17:00 LST) ozone for coastal, urban, and mountain areas between 2004 and
2019.

(a)











(b)











(c)










Fig. 3. Near-surface weather charts obtained from NCEP GFS data. Gray area
represents cloud area according to a Himawari satellite infrared image. (a) 00:00
UTC, 15 July; (b) 00:00 UTC, 16 July; and (c) 00:00 UTC, 17 July.




(a)                                          (b)
(c)




Fig. 4. Morning sounding launched at 00:00 UTC at station 46734 (located at MG in

Fig. 1a) (a) 15 July (b) 16 July, and (c) 17 July.


(a)                                          (b)




(c)


...

Fig. 5. Hourly variation of observed (red) and simulated (blue) values for (a) wind, (b)
PM$_{2.5}$, and (c) daytime (08:00-17:00LST) ozone between July 12 and 18, 2018, for
the coastal, urban, and mountain stations as well as for the two island stations,
Kinmen (KM) and Magong (MG).



(a)                          (b)                          (c)

(d)                          (e)                          (f)



Figure 6 Observed PM2.5 concentration and wind recorded in Taiwan (a) at 08:00
LST (b) 12:00 LST (c) 14:00 LST (d) 16:00 LST (e) 20:00 LST, 17 July (f) 00:00
LST, 18 July, 2018.










Figure 7 Observed ozone concentration (ppb) and wind recorded in Taiwan at (a)
08:00 LST (b) 10:00 LST (c) 12:00 LST (d) 14:00 LST (e) 16:00 LST (f) 18:00 LST
(g) 20:00 LST (h) 22:00 LST, 17 July, 2018.



(a)          (b)          (c)


(d)          (e)          (f)
Figure 8 Simulated streamline and PM$_{2.5}$ concentration (µg/m$^3$) in Taiwan (a) at 08:00
LST (b) 12:00 LST (c) 14:00 LST (d) 16:00 LST (e) 20:00 LST, 17 July (f) 00:00
LST, 18 July, 2018.



(a)

(b)

(c)

(d)

(e)

(f)

(g)

(h)

(i)


Figure 9 Simulated streamline and ozone concentration (ppb) in Taiwan at (a) 04:00
LST (b)08:00LST (c) 10:00 LST (d) 12:00 LST (e) 14:00 LST (f) 16:00 LST,
(g)18:00 (h) 20:00 LST, (i) 22:00 LST, 17 July, 2018 .





(a)                              (b)                              (c)


(d)                              (e)                              (f)

(g)                              (h)                              (i)


Figure 10 (a) The geographic location of study area in central Taiwan and the location
of NW-SE cross section AA'. Wind field distribution and PM$_{2.5}$ concentration (unit:
μg/m$^3$) along the northwest–southeast cross section at (b) 05:00 LST (c) 07:00 LST
(d) 10:00 LST (e) 12:00 LST (f) 14:00 LST (g) 16:00 LST (h) 18:00 LST (i) 20:00
LST, 17 July, 2018.

(a)              (b)              (c)
(d)              (e)              (f)
(g)              (h)              (i)




Figure 11 Wind field distribution and ozone concentration (unit: ppb) along the
northwest–southeast cross section AA' in Figure 10a, at (a) 04:00 LST (b) 08:00
LST (c) 10:00 LST (d) 12:00 LST (e) 14:00 LST (f) 16:00 LST (g) 18:00 LST (h)
20:00 LST (i) 23:00 LST










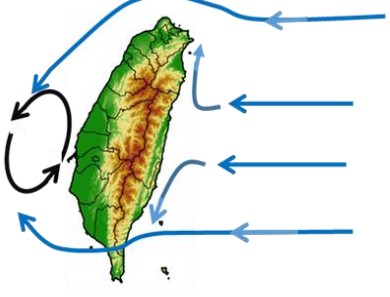

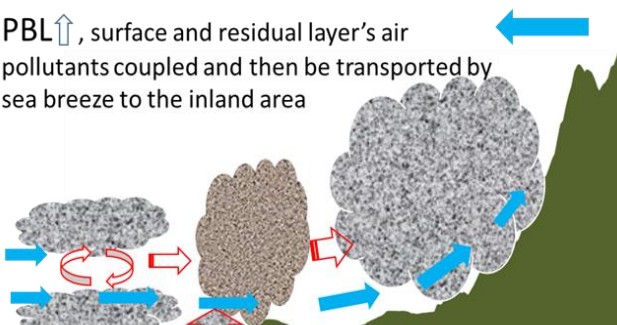




Figure 12 Schematic of the processes of air quality deterioration episode associated
with typhoon over Taiwan's western plain





