# Peer review of "Air quality deterioration episode associated with typhoon over the"

_Atmospheric Chemistry and Physics, 2021_

## Author Response (AR1)

**Response to reviewer#1**

Major revision:

> 1. …..my primary concern is about how the interesting results here are to be generalized to other locations of the world. Such a statement is better included.

R: Thank you for the comments and constructive suggestions. We included this suggestion in the revision. **(L89-L103)**

The interactions between ambient flow and topography resulted in stable weather conditions and air pollutants accumulation in the low boundary are common all over the world. Actually, according to the obstacle's scale, it could occur in Plateau (Ning et al. 2019 ), mountain (Lai and Lin et al. 2020) and even buildings environment (Theurer W., 1999 ) as the airflow interacted with them. For example, Wallace et al. (2010) investigated the spatial and topographic effects of temperature inversions on air quality in the industrial city of Hamilton, located at the western tip of Lake Ontario, Canada. Topographically constrained wind flows and frequent temperature inversions occurred at Los Angeles, California (Lu and Turco, 1995), the Highveld Plateau industrial region in South Africa (Jury and Tosen, 2004), and Perth, Australia (Pitts and Lyons, 1988). Valverde et al. (2016) studied air pollution in Europe and found that the dispersion and transfer of air pollutants are affected by topographic features and weather patterns. Ning et al.(2019) presented synergistic effects of synoptic weather patterns low trough, low vortex and topographic on air quality over the Sichuan Basin of China.

Once a tropical cyclone existed and usually enhanced the interaction between airflow with the mountain and thus advise to the air pollutants diffusion. Some papers that discussed tropical cyclone impacts on the air quality over the western North Pacific surrounding region such as southern China (e.g. Huang et al., 2006, 2005 ), **HongKong** (e.g. Zhang et al. 2013, Yang et al. 2019), **Korea** (Park et al. 2019) and **Japan (**Pan et al. 2016**).** Park et al. (2019) found that TCs in the South China Sea can enhance the PM10 concentration over South Korea through poleward-propagating Rossby waves. Generally, air conditions become stable right before a TC reaches a region because of the dominant descending motion surrounding the TC (Feng et al. 2007; Chow et al. 2018; Liu et al. 2018).

**References:**

Chow, E. C. H., Li, R. C. Y., and Zhou, W.: Influence of tropical cyclones on Hong Kong air quality,

Adv. Atmos. Sci., 35, 1177–1188, https://doi.org/10.1007/s00376-018-7225-4, 2018.

Feng, Y., Wang, A., Wu, D., and Xu, X.: The influence of tropical cyclone Melor on PM10 concentrations during an aerosol episode over the Pearl River Delta region of China: numerical modeling versus observational analysis, Atmos. Environ., 41, 4349–4365, https://doi.org/10.1016/j.atmosenv.2007.01.055, 2007.

Huang, J. P., Fung, J. C. H., Lau, A. K. H., and Qin, Y.: Numerical simulation and process analysis of typhoon-related ozone episodes in Hong Kong, J. Geophys. Res., 101, https://doi.org/10.1029/2004JD004914, 2005.

Huang, J. P., Fung, J. C. H., and Lau, A. K. H.: Integrated processes analysis and systematic meteorological classification of ozone episodes in Hong Kong, J. Geophys. Res., 111, https://doi.org/10.1029/2005JD007012, 2006.

Jury MR, Tosen GR. Characteristics of the winter boundary layer over the African Plateau: 26°S. Boundary-Layer Meteorol;49:53–76, doi:10.1007/BF00116405, 2004.

Lai H.C., Lin M.C., Characteristics of the upstream flow patterns during PM2.5 pollution events over a complex island topography, Atmos Environ;227:117418, 2020.

Liu,W., Han Y., Yin Y., Duan J., Gong J., Liu Z., Xu W., An aerosol air pollution episode affected by binary typhoons in east and central China, Atmos. Pollution Res., 9, 634-642, 2018.

Lu R, Turco RP. Air pollutant transport in a coastal environment. II: Three-dimensional simulations over Los Angeles Basin. Atmos Environ;29:1499–518, 1995.

Ning G., Yim S.H.L, Wang S., Duan B., Nie C., Yang X., Wang J., Shang K., Synergistic effects of synoptic weather patterns and topography on air quality: a case of the Sichuan Basin of China., 53, 6729-6744, 2019.

Pan, X., Uno, I., Hara, Y., Osada, K., Yamamoto, S., Wang, Z., Sugimoto, N., Kobayashi, H., and Wang, Z.: Polarization properties of aerosol particles over western Japan: classification, seasonal variation, and implications for air quality, Atmos. Chem. Phys., 16, 9863–9873, https://doi.org/10.5194/acp-16-9863-2016, 2016.

Park D.R., Ho C. H., Kim D., Kang N. Y., Han. Y., OH H. R., Tropical Cyclone as a Possible Remote Controller of Air Quality over South Korea through Poleward-Propagating Rossby Waves, J. Meteor. Soc. Japan, 58, 2523-2530, 2019.

Pitts O, Lyons TJ. The influence of topography on Perth radiosonde observations. Aust Meteorol Mag 1988;36:17–23.

Theurer W., Typical building arrangements for urban pollution modelling. Atmos. Environ., 33, 4057-4066, 1999.

Valverde V., M.T. Pay, J.M. Baldasano, A model-based analysis of SO2 and NO2 dynamics from coal-fired power plants under representative synoptic circulation types over the Iberian Peninsula. Sci. Total Environ., 541, 701-713, 2016.

Wallace J., Corr D., Kanaroglou P., Topographic and spatial impacts of temperature inversions on air quality using mobile air pollution surveys.    Science of the Total Environment 408, 5086–5096,

2010.

Yang, Y., Yim, S. H. L., Haywood, J.,Osborne, M., Chan, J. C. S., Zeng, Z., Cheng, J. C. H.,
Characteristics of heavy particulate matter pollutionevents over Hong Kong and theirrelationships
with vertical wind profiles using high-time-resolution Dopplerlidar measurements. Journal of
Geophysical Research: Atmospheres,124, 9609–9623 http://dx.doi.org/10.1029/2019JD031140,
2019.

Zhang, Y., Mao, H. T., Ding, A. J., Zhou, D. R., and Fu, C. B.: Impact of synoptic weather patterns on
spatio-temporal variation in surface O3 levels in Hong Kong during 1999–2011, Atmos. Environ.,
73, 41–50, https://doi.org/10.1016/j.atmosenv.2013.02.047, 2013.

2. Second, there are several parts where additional justification and explanation are necessary. For example, description on observations is weak, for both TEPA monitoring and campaign observations (for example, in-situ measurement techniques, sampling flow rates, and ion analysis). Also, the roles of biogenic VOCs emitted from the forests near the mountain regions should be discussed.

R: Thank you for the constructive suggestions. We further provided the information about TEPA monitoring and campaign observations discussed and responosed in detail in the specific questions as following in specific comments (Q#1 and #2).

Specific comments:

1, line 113-114: Specify measurement techniques for PM2.5 and O3, whose data are heavily used.

R: The instruments of the measurement of $PM_{10}$ and $PM_{2.5}$ are METONE_BAM1020. Two reactive gases, ozone (O3) and sulfur dioxide (SO2), were measured in parallel with the aerosol measurements. A non-dispersive ultraviolet photometer (ML9810, Ecotech, Australia) and an ultraviolet fluorescence spectrometer (ML9850, Ecotech, Australia) are used to measure O3 and SO2 concentrations, respectively. (https://airtw.epa.gov.tw/CHT/EnvMonitoring/Central/Tools.aspx) Table 1R lists the measurement frequency and the methodologies used in this study. **(L161-167)**

Table 1R  methods and instruments used in this study

| Species | Method | Instrument | Frequency |
|---|---|---|---|
| $PM_{10}$, $PM_{2.5}$ | Beta ray attenuation | Met-One BAM-1020 | Hourly |
| $O_3$ | Nondispersive ultraviolet (UV) photometry | Ecotech 9810 | Hourly |

| NO$_x$ | Chemiluminescene | Ecotech 9841 | Hourly |
| SO$_2$ | UV fluorescence detection | Ecotech 9850 | Hourly |
| CO | Non-dispersion infrared analysis | HORIBA_APMA360 | Hourly |

2.  Lines 135-136: Were high-volume air samplers used for sampling and subsequent chemical analysis? What was the flow rate? Measurement methods for sulfate, nitrate, ammonium, and EC/OC should be described. Maybe the sampling periods are 11 h (not 12 h)?

R: The sampling period of each sample was 11 h; daytime samples were collected from 08:00 to 19:00 LST, whereas nighttime sampling was conducted from 20:00 LST to 07:00 LST **(L152-154)**.

The sampler was located in an open area and the sampling flow rate was set to 6.7 L/min for each channel. The filter samples were analyzed for water-soluble ions (Ca$^{2+}$, Mg$^{2+}$, Na$^+$, NH$_4^+$, K$^+$, SO$_4^{2-}$, NO$_3^-$ and Cl$^-$) via ion chromatography (Dionex ICS 1000, Thermo Scientific). The nylon filter was deployed as the backup filter to correct negative artifact for NO3-, whereas the MgO denuder was deployed to remove the positive artifact from gaseous HNO3.   Organic Carbon (OC) and Elemental Carbon (EC) were measured by a thermal/optical carbon analyzer (DRI, 2001A, Atmoslytic Inc.), following the IMPROVE thermo-optical reflectance (TOR) protocol (Chow et al., 2001). The sampling period is 11 hours. The text has been amended in the revision. **(L157-161)**

3. What is this number (46734) for?

R: It is World Meteorological Organization (WMO) station number code, i.e. "46734" is standing for Magong sounding station (MG, Figure 1a). **(L168-169)**

4. Line 217: For the buildup of OC, what are the roles of biogenic VOCs emitted from the forests near the mountain regions? I thought model simulations could tell the importance.

R: Thank you for the comments. As presented in Lee (et al. 2019), the mass contribution of secondary organic carbon (SOC) to PM2.5 peaked in summer (13.2%) over western Taiwan, inferring the importance of enhanced photo-oxidation reactions in SOC formation due to the high solar radiation in the summertime.

Figure R1a showed the simulated spatial distribution of the OC on the event day (17 July) while the simulation difference without biogenic VOCs emitted from forests as shown in Figure R1b. It was estimated the biogenic VOCs contribution about 10-20% (2-3 ug/m$^3$) in central Taiwan.

Ref:

Lee, C. S. L., Chou, C. C.-K., Cheung, H. C., Tsai, C.-Y., Huang, W.-R., et al.: Seasonal variation of chemical characteristics of fine particulate matter at a high-elevation subtropical forest in East Asia, Environ. Pollut., 246, 668–677, https://doi.org/10.1016/j.envpol.2018.11.033, 2019

[Figure]

Figure R1 (a) Spatial distribution of simulation OC on the event day 17 July 2018 (b) The simulation spatial distribution of OC for the difference between with and without biogenic VOCs on the event day 17 July 2018.

5. Lines 220-222: Chemical aging was not important for the buildup of sulfate?

R: Chemical aging should play a role in this study. As shown in the manuscript in Table 1, the concentration of sulfate at coastal, urban, and mountain are 4.5, 4.6, and 4.8 ug/m3, respectively. In other words, the mountain site is equal or even minor higher than the others due to the land-sea breeze and the effective formation of nss-SO$_4^{2-}$ primarily from SO$_2$ during the mountain upslope transport. Similar results were also shown in Lee et al. (2019), they presented notable high SOR at the site and diurnal variations of O3 and SO2 coincided with each other.

Ref:

Lee, C. S. L., Chou, C. C.-K., Cheung, H. C., Tsai, C.-Y., Huang, W.-R., et al.: Seasonal variation of chemical characteristics of fine particulate matter at a high-elevation subtropical forest in East Asia, Environ. Pollut., 246, 668–677, https://doi.org/10.1016/j.envpol.2018.11.033, 2019

6. Line 229: Which species are emitted from agriculture?

R: The formation of $NO_3^-$ could be partly from neutralization of NOx gases, however, it could be attributed to other sources of $NO_3^-$, such as from fertilizers (e.g. $Ca(NO_3)_2$) and organic nitrates (Perring et al., 2009; Day et al., 2010). In western Taiwan, a high fraction of $NO_3^-$ (23.5%) was observed in spring which contributed comparably with the nss-$SO_4^{2-}$ (23.2%) to the $PM_{2.5}$ mass (Lee et al. 2019). A possible reason for this is the release of nitrogen oxides due to fertilization in adjacent agricultural counties.

Ref:

Day, D.A., Liu, S., Russell, L.M., Ziemann, P.J., 2010. Organonitrate group concentrations in submicron particles with high nitrate and organic fractions in coastal southern California. Atmos. Environ. 44 (16), 1970e1979.

Perring, A.E., Wisthaler, A., Graus, M., Wooldridge, P.J., Lockwood, A.L., Mielke, L.H., Shepson, P.B., Hansel, A., Cohen, R.C., 2009. A product study of the iso-preneþNO3 reaction. Atmos. Chem. Phys. 9, 4945-4956.

7. Line 267. Bifurcated air flows meeting in the western Taiwan may form a kind of convergence. Why does the convergence result in subsidence rather than updraft?

R: The central mountain range (CMR) is oriented in an NNE-SSW direction in Taiwan, more than 200 km in length, and with an average terrain elevation of about 2000 m (Fig 1b). In this study, the atmospheric condition is quite stable due to typhoon circulation enhanced the easterly flow and subsidence is significant in the lee side of the CMR over western Taiwan. The morning sounding (Fig 4) at MG station significantly presented the depth of the inversion layer that existed below 850 hPa. The confluent flow due to bifurcated airflow at the lee side was relatively weak and coupled with a sea breeze at near-surface during daytime (Fig 8 a-c). Thus, the stable weather conditions resulted in subsidence that dominated over western Taiwan.

8. Line 285: Do the authors mean change over time?

   R: Yes, In Fig 5a, it is an hourly data variation of the wind field and PM2.5 concentration.

9. Line 306-307: Titration of O3 by freshly emitted NO might be another reason for the low ozone near the sources?

R: Thank you for the comment. Yes, other than the meteorological condition, the titration of O3 by freshly emitted NO might be another reason for the low ozone near the sources.

10. Line 335-336: Are the correlation coefficients as high as 0.72 and 0.81 from hourly values?

R: Yes, the hourly data was employed for the correlation coefficient calculation.

11. Lines 398-400: Here mentioning a branch flowing over the ocean to the south? Please clarify.

R: Text has been amended. The high-ozone plume was associated with the lee vortex circulation over the Taiwan Strait and existed during the nighttime and early morning **(L430-431)**

12. Table 2: July 15, 2018 is Sunday. Any emission change on this day?

R: Thank you for the comment. In western Taiwan, the plain area is narrow. The major emission sources are including power plants, traffic, and industry in central Taiwan. Figure 5 presented the time series of PM2.5 and O3. Comparing to the other days and the locations at the coast, urban, and mountain, the variation trends are similar at these sites. We believe the holiday effect is minor and the meteorological conditions are the dominant factors.

13. Figure 4. Add explanation of the colored lines in the caption or legend.

   R: Thank you for the suggestion. Red line represents vertical profile air temperature and blue is dewpoint temperature. Texts in Figure 4 have been amended in the revision. **(L866-867)**

14. Figure 5a. Land-sea breeze is well resolved with the model with a resolution of 3 km. This should be better highlighted, with some comparison

to coarser resolution model cases. Are there any reasons why the simulated wind speeds are often higher? And any potential influence of the stronger wind on the model results of O3 and PM2.5?

R: Thank you for the suggestions. We further emphasized the well simulation on the land-sea breeze in the revision. We obtained the meteorological initial and boundary conditions for WRF-Chem from the National Center for Environmental Prediction (NCEP) Operational Global Forecast system $0.25° \times 0.25°$ data sets at 6-h intervals **(L187-189)**. In other words, its resolution is 25 km. Furthermore, the topographic and terrain height will be smoothed especially for the small island Kinmen.   As shown in Figure 5a, the observed and simulated $PM_{2.5}$ and $O_3$ are all low due to strong wind during the episode days. Although the simulated wind speed is stronger than the observed at KM station, it doesn't influence our explanation on this phenomenon.

15.     Figure 6. Mention unit of the PM2.5 mass concentrations in the caption.

R: Text in Figure 6 has been amended in the revision. **(L897)**

16.     Figures 8 and 9: Which altitude are the shown streamlines for?

R: It is first layer of the simulated streamlines (30 m height). Texts have been added in the captions in Figures 8 and 9. **(L920 and L927)**

**Response to Reviewer#2**

Major Comments:

   In this manuscript, the authors presented "Air quality deterioration episode associated with typhoon over the complex topographic environment in central Taiwan". The schematic summary in Figure 12 is interesting and clearly presented the diurnal process during the episode. This manuscript is well organized and is helpful to understand the interactions between ambient flow, land-sea breeze, boundary layer development with topographic in Taiwan and other regions.    I suggest it be accepted after considering the following comments.

   1. Mechanism: It seems the easterly flows interacted with the central mountain range(CMR) is an important process and easily to formation air quality over western Taiwan. Although typhoon circulations provided a stronger easterly flow, I am just wondering if a typhoon is not a necessary condition. In other words, perhaps other weather conditions also will formation such an air quality deterioration, if so, could you also mention it in your article?

   R: Thank you for the suggestion. As mentioned in the manuscript, the interaction between the easterly flow and Taiwan's Central Mountain Range (CMR) resulted in stable weather conditions and weak wind speed in western Taiwan. During summer monsoon, the prevailing wind is southwesterly over Taiwan. However, the abnormal enhanced easterly flow usually occurs as the typhoon track between Taiwan and Luzon island of the Philippines. Actually, the easterly flow is commonly prevailing during the winter monsoon, i.e., the northeasterly or easterly continental outflow from autumn to springtime in this region (Figure R1.1). That's the reason why air quality episode frequently occurred over western Taiwan during winter monsoon ( e.g. Lin et al. 2004, 2005, and 2012a,b) **(L79-82).**

[Figure]

   Figure R1.1 Schematic of the relationship between the movement of continental highs and associated wind direction around Taiwan

2. The authors presented the sources of meteorological initial and boundary conditions for WRF-chem model and simulated well for the pollutants' variations in Fig. 5. Which anthropogenic emission inventory in Taiwan do you use? the information on emission inventories used in WRF-Chem simulation is not introduced.

R: The anthropogenic emissions were obtained from the air-pollutant monitoring database of the Environmental Protection Agency, Taiwan. Its emission inventory system, called Taiwan Emission Data System (TEDS). The TEDS version in this study is V9.0 (2013) and contains data on eight primary atmospheric pollutants, CO, NO, NO2, NOx, O3, PM10, PM2.5, and SO2. Pollution emissions of all activities at national and regional levels as well as from companies/institutions and other emission sources both stationary and mobile are covered by the TEDS. **(L203-207)**

3. The authors presented the aerosol composition analysis over western Taiwan and showed the differences in coastal, urban, and mountain sites. I am just curious what kind of instruments (models, ..) you used? I suggest you give more information.

R: The filter samples were analyzed for water-soluble ions ($Ca^{2+}$, $Mg^{2+}$, $Na^+$, $NH_4^+$, $K^+$, $SO_4^{2-}$, $NO_3^-$ and $Cl^-$) via ion chromatography (Dionex ICS 1000, Thermo Scientific). The nylon filter was deployed as the backup filter to correct negative artifact for NO3-, whereas the MgO denuder was deployed to remove the positive artifact from gaseous HNO3. Organic Carbon (OC) and Elemental Carbon (EC) were measured by a thermal/optical carbon analyzer (DRI, 2001A, Atmoslytic Inc.), following the IMPROVE thermo-optical reflectance (TOR) protocol (Chow et al., 2001).
Furthermore, the instruments of the measurement of PM10 and PM2.5 from Taiwan EAP are METONE_BAM1020. **(L157-167)**

4. L135-137 The sampling period of each sample was 12 h; daytime samples were collected from 08:00 to 19:00 LST, whereas nighttime sampling was conducted from 20:00 LST to 07:00 LST. Please check the sampling period, because nighttime is only 11 h.

R: The sampling period is 11 hours. The text has been amended in the revision **( L152-154 ).**

5. Line 181: Figure 2a-c, the monthly mean results for PM10, PM2.5 and daytime ozone. What's the analysis period? It seems to me has a typo error for the starting in 1994, Please double check it.

R: The text has been amended ( L212). The study period is between 2004 and 2019.

6. Line 294-295, 18:00 LT........,; Line 487 " daytime ozone (08:00-17:00 LT); Please consistently present the abbreviation of local standard time (LST) or LT,

R: The text has been amended. (L326; L520).

7. Line 318, Is it "Maoli county",   or " Miaoli county" ?,   In Figure 2 b, it is "Miaoli" please double check it.

R: It should be "Miaoli". The text has been amended in this revision ( L349).

8. Summary: It is wordy, loss of focus, and needs to be more concise.

R:   Thank you for your suggestion. We re-organized the summary section in more concise as shown in this revision (L518-558)

9. References: Line 83:    Lin Y. L. (1993) was not shown in the reference list. ; Line 48 and Line 601: it should be Huang et al. (2021). Line 635:" Atmos. Environ., 41, 3684–3701, https://doi.org/10.1016/j.atmosenv.2006.12.050, 2007b" , it should be 2007.

R: Thank you for the suggestion. The text has been amended (L48; L663). We also added the reference, Lin Y. L. (1993), in the list (L677-678)